# Predictors of SIV recrudescence following antiretroviral treatment interruption

**Mykola Pinkevych[1], Christine M Fennessey[2], Deborah Cromer[1], Carolyn Reid[2], Charles M Trubey[2], Jeffrey D Lifson[2], Brandon F Keele[2]\*, Miles P Davenport[1]\***

[1]Infection Analytics Program, Kirby Institute for Infection and Immunity, UNSW Australia, Sydney, Australia; [2]AIDS and Cancer Virus Program, Frederick National Laboratory for Cancer Research, Frederick, United States

**Abstract** There is currently a need for proxy measures of the HIV rebound competent reservoir (RCR) that can predict viral rebound after combined antiretroviral treatment (cART) interruption. In this study, macaques infected with a barcoded SIVmac239 virus received cART beginning between 4- and 27 days post-infection, leading to the establishment of different levels of viral dissemination and persistence. Later treatment initiation led to higher SIV DNA levels maintained during treatment, which was significantly associated with an increased frequency of SIV reactivation and production of progeny capable of causing rebound viremia following treatment interruption. However, a 100-fold increase in SIV DNA in PBMCs was associated with only a 2-fold increase in the frequency of reactivation. These data suggest that the RCR can be established soon after infection, and that a large fraction of persistent viral DNA that accumulates after this time makes relatively little contribution to viral rebound.

## Introduction

Current treatment for HIV infection requires an individual to undergo lifelong daily cART administration, as interruptions in dosing usually result in the rapid recrudescence of virus. This viral recrudescence is thought to result from periodic activation of viral production from cells, including latently infected cells (*Pinkevych et al., 2015*; *Fennessey et al., 2017*), allowing the virus to spread to uninfected target cells no longer protected from infection by cART. A variety of approaches are currently being studied to reduce the RCR or the frequency of activation leading to virus production, in order to produce anti-retroviral free remission in treated patients (*Deeks et al., 2016*; *Davenport et al., 2019*). A variety of assays have been developed to measure HIV DNA (total or integrated virus, intactness of viral genomes, or replication competence in outgrowth assays), levels of HIV-RNA production (spliced or unspliced), or the ability to drive viral production through in vitro or in vivo stimulation of cells (*Eriksson et al., 2013*; *Procopio et al., 2015*; *Finzi et al., 1999*; *Lorenzi et al., 2016*; *Metcalf Pate et al., 2015*; *Barton and Palmer, 2016*; *Bruner et al., 2015*). These measures usually demonstrate a very wide range of values between individuals, varying between 100-fold and 1,000-fold for different individuals, and different measures are often poorly correlated (*Eriksson et al., 2013*). Moreover, although these are often described as measures of the 'viral reservoir', their contribution to the actual post-ART viral recrudescence is unclear.

While complete eradication of the HIV RCR would prevent viral recrudescence, greatly decreasing the frequency of cells producing infectious virus may provide prolonged remission and the possibility of functional 'cure' (*Davenport et al., 2019*; *Cromer et al., 2017*; *Hill et al., 2016*). In human studies, time-to-detectable viremia can be used as a rough estimate of the size of the RCR in a cohort of individuals (*Pinkevych et al., 2015*) and has been used to compare the size of the RCR between groups (*Bar et al., 2016*; *Colby et al., 2018*; *Li et al., 2015*). However, this approach has very limited statistical power, due in part to variability of the exact time first detectable viremia occurs and

\*For correspondence:
keelebf@mail.nih.gov (BFK);
M.Davenport@unsw.edu.au (MPD)

**Competing interests:** The authors declare that no competing interests exist.

**eLife digest** Several drugs are available to control HIV, but they do not completely eliminate the virus from the body. Instead, these treatments stop the virus from multiplying, but unless a person is treated very soon after infection, inactive HIV can hide inside cells, and the infection is not completely cleared. Once treatment stops, the inactive virus starts to multiply, reaching pre-treatment levels within weeks. This means that infected individuals must continue treatment for life, or the virus will return and may cause disease. To prevent this, scientists are trying to find a way to eliminate HIV from the body, permanently curing the disease.

Testing HIV drugs is difficult because there is no simple way to determine if all the inactive virus has been removed. One way to test whether a person is cured is to stop treatment, and see if the virus comes back. An alternative method is to measure the amount of inactive HIV present in blood cells. However, Pinkevych et al. have now shown that levels of inactive HIV in the blood may not be a good predictor of whether HIV levels will rebound after treatment.

In the experiments, Pinkevych et al. infected macaques with a monkey version of HIV called simian immunodeficiency virus, or SIV for short. The virus was genetically modified to have a 'genetic barcode' that allowed researchers to track thousands of individual strains of virus simultaneously . The macaques were then treated with a combination of drugs starting either 4 or 27 days after infection, and then the treatment was stopped to check how rapidly strains of the virus would re-emerge.

Although the virus rebounded in both groups, during treatment the inactive virus was often undetectable in the blood from the group treated four days after infection , suggesting that the virus may be hiding elsewhere. In the group treated 27 days after infection, more blood cells had inactive virus and reactivation was more frequent. However, the amount of inactive virus in the blood did not directly predict the frequency of reactivation: a 100-fold increase in viral levels only led to reactivation being twice as frequent. This suggests that the amount of inactive virus in the blood is not a good read-out for whether the virus will come back.

These experiments suggest that inactive virus hides in cells quickly, since treatment just four days after infection failed to eliminate the virus completely, and that it must hide in cells other than blood cells. Evidence from humans suggests something similar may occur in HIV infection. Further studies may help identify which cells harbor the inactive virus and contribute to infections re-emerging after treatment stops.

wide confidence intervals within individual groups. It has been estimated that groups of >100 individuals may be required to convincingly demonstrate even quite substantial changes in the reservoir size with treatment (*Pinkevych et al., 2015*; *Moore et al., 2019*). Therefore, the efficacy of these interventions is often quantified by surrogate measures such as the levels of various forms of HIV DNA, cell-associated HIV RNA, or the ability of latent cells to be reactivated ex-vivo (*Rasmussen et al., 2014*; *Søgaard et al., 2015*; *Elliott et al., 2014*). Understanding how these surrogate measures relate to the frequency of HIV reactivation and the resulting duration of viral rebound free remission after treatment interruption would greatly facilitate the accurate evaluation of novel therapies to reduce the RCR (*Eriksson et al., 2013*; *Sharaf and Li, 2017*).

The frequency of individual infected cells able to activate and produce sufficient viral progeny to allow for a spreading infection is postulated to be determined by a number of factors, including; (i) the number of infected cells harbouring HIV DNA, (ii) the fraction of that DNA representing replication-competent proviruses, (iii) the viral integration site and level of cellular activation and epigenetic and transcriptional states of individual proviruses (which together may help determine the per cell frequency of reactivation of virus; *Hill, 2017*; *Yukl et al., 2018*), and (iv) the level of immune control of viral reactivation. A number of studies have reported that levels of integrated HIV DNA are a poor predictor of time-to-recrudescence of virus. *Williams et al. (2014)* investigated a cohort of 154 individuals who initiated treatment in the chronic phase of infection and underwent treatment interruption, and found that integrated HIV DNA was not a significant predictor of time-to-recrudescence (defined as time to reach 50 copies/ml of virus in plasma; *Williams et al., 2014*). In a study by *Calin et al. (2016)* , patients selected for having very low HIV DNA levels failed to show a delayed

time-to-rebound viremia. A recent study by *Colby et al. (2018)* showing time-to-recrudescence of virus in patients with very low HIV DNA who were treated within weeks of infection (in Fiebig stage 1) suggests relatively little change in the frequency of HIV reactivation from latency compared to cohorts treated in chronic infection. However, other studies have suggested that early ART treatment in combination with lower levels of HIV DNA at interruption may be associated with delayed time-to-HIV-relapse (*Li et al., 2015*; *Williams et al., 2014*; *Assoumou et al., 2015*; *Wen et al., 2018*; *Steingrover et al., 2008*). Therefore, the relationship between surrogate measures of HIV 'reservoir' (in peripheral blood) and the later time-to-viral-rebound remains unclear. Moreover, in studies of the efficacy of candidate latency reducing strategies, it is unclear if changes measured in peripheral blood will be associated with meaningful changes in post-treatment time to viral rebound (*Petravic et al., 2017*).

In this study, we used a macaque model of AIDS virus infection, employing the barcoded SIVmac239M virus to investigate the dynamics of SIV rebound after treatment interruption, and assess what host factors might correlate with or predict cellular activation and subsequent viral rebound. The use of the barcoded virus allows estimation of the frequency of SIV reactivation from latency in individual animals (*Fennessey et al., 2017*). We studied viral reactivation rates in macaques in which cART was initiated at different times after infection (between day 4 and day 27), allowing for differing extents of viral dissemination reflected by different levels of viral DNA (vDNA) (spanning two logs) in PBMC. SIV DNA levels decayed under treatment and when cART was stopped after 43–68 weeks of treatment, we were able to estimate the frequency of SIV reactivation following treatment interruption. While SIV DNA level in PBMC just before treatment interruption was a predictor of SIV reactivation, this relationship appeared extremely 'flat'. That is, an $\approx 100$ fold increase in SIV DNA levels was associated with only a two-fold difference in reactivation rates during a analytic treatment interruption (ATI). We also investigated a variety of other factors that could plausibly affect the relationship between vDNA levels and the frequency of reactivation from latency. This work suggests that the rebound competent reservoir may be saturable and can be seeded early in the course of infection Subsequent increases in viral DNA due to a delay in therapy only marginally alters the dynamics of post-ATI rebound.

## Results

### Measuring the frequency of SIV reactivation from latency

We previously developed a barcoded SIVmac239 virus, designated SIVmac239M, containing $\approx 10{,}000$ clonotypes, present at approximately equal proportions (*Fennessey et al., 2017*). Infection of macaques with a suitable inoculum of this virus creates an essentially isogenic and phenotypically equivalent population of viruses containing diverse barcodes in both the circulating plasma virus and the SIV DNA of infected cells. cART leads to prolonged suppression of virus in this model, and treatment interruption is followed by rapid viral rebound. We previously reported studies of a cohort of rhesus macaques, infected intravenously with $2.2 \times 10^5$ IU of SIVmac239M, and treated with cART initiated on day four post-infection for 300–478 days followed by treatment interruption (*Fennessey et al., 2017*) (*Figure 1a*, n = 6). In the present study, we extended this work to investigate macaques in which cART was initiated on day 10 (n = 4) or day 27 (n = 5) post-infection (hereafter referred to as 'day 10 treated' and 'day 27 treated' respectively). Animals were treated for between 310 and 476 days, followed by treatment interruption. The plasma viral loads for the nine animals treated beginning on days 10 and 27 are shown (*Figure 1c,e*). Following treatment interruption, we measured the growth rate of total plasma viral RNA and calculated the proportional contribution of individual SIVmac239M barcode clonotypes (obtained using high throughput sequencing of rebound virus) to the total rebound viremia (*Figure 1b,d,f*). We then used a mathematical modelling approach to estimate the average frequency of reactivation from latency in individual animals (*Pinkevych et al., 2015*; *Fennessey et al., 2017*)(*Figure 1g–i*).

For the four macaques treated starting on day 10 post-infection, the estimated frequency of reactivation from latency varied between 1.63 and 2.94 reactivations per day (mean = 2.29, SD = 0.47, n = 4). For animals treated beginning on day 27, the frequency varied between 0.75 and 3.10 reactivations per day (mean 1.33, SD = 0.90, n = 5) and was not significantly different from day 10 animals p=0.19 (Mann Whitney). The frequency of reactivation of animals treated on or after peak infection

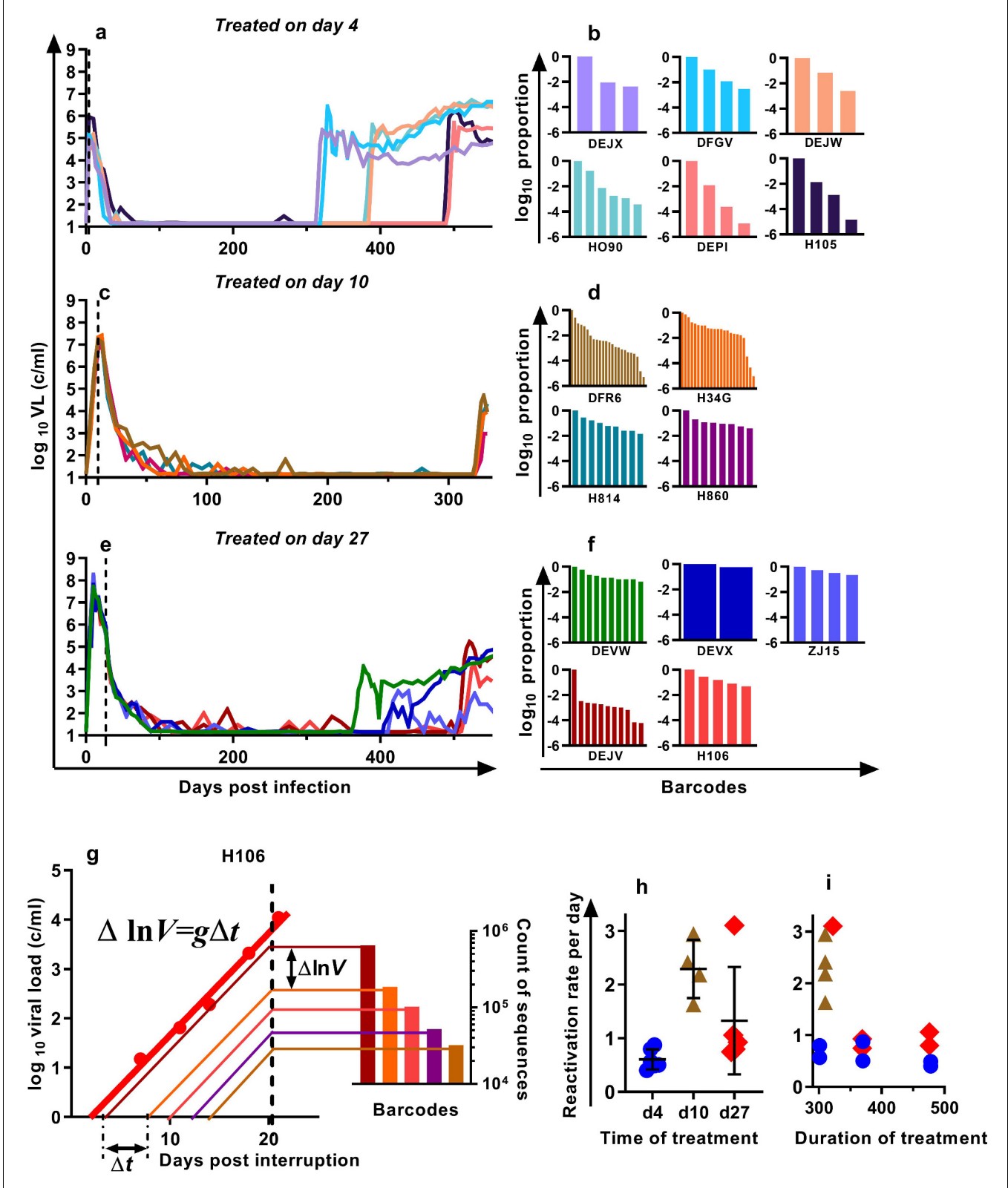

**Figure 1.** Viral load and frequency of reactivation after treatment interruption. The viral load in individual animals infected with SIVmac239M and (a) treated on day four and interrupted on days 300 (n = 2), 370 (n = 2), or 478 (n = 2) and reproduced from *Fennessey et al. (2017)*, or (c) treated day 10 and interrupted on day 320 (n = 4), or (e) treated on day 27, and interrupted on days 349 (n = 1), 396 (n = 2), or 503 (n = 2). Vertical dashed line is the time of treatment. (b,d,f) Frequency of barcode clonotypes identified by high throughput sequencing of plasma virus in individual monkeys during

*Figure 1 continued on next page*

*Figure 1 continued*

rebound. (g) Schematic of the method for estimation of the frequency of reactivation from latency, detailed in *Fennessey et al. (2017)*. Red dots – measurements of total viral load. Thick red line is the trajectory of growth of total viral load. Coloured thin lines are the theoretical trajectories of growth of individual clonotypes based on the proportion of individual barcode sequences in the rebound plasma (bar graph at the right-hand side). Time between reactivations Δt is proportional to difference between logarithms of the frequency of barcodes. (h) The frequency of reactivation for the three cohorts initiating treatment on different days post-infection (bars indicate mean and SD) and (i) treated for different lengths of time.

The online version of this article includes the following source data for figure 1:

**Source data 1.** Viral kinetics, clonotype proportions, and reactivation frequencies for SIVmac239M infected animals.

(pooled day 10 and day 27 groups) was significantly higher (Mann Whitney's p=0.0028) than the frequency of reactivation previously measured for animals treated at day four post-infection, which were between 0.40 and 0.87 per day (mean = 0.61, SD = 0.17, n = 6) (*Figure 1h*). Longer duration of treatment also appeared to be associated with declining frequency of reactivation (half-life 216 days), although this was not significant (p=0.064, linear mixed effects (LME) model) (*Figure 1i*).

## SIV DNA in PBMC and frequency of reactivation

In order to understand the association between peripheral blood virologic measurements and the estimated frequency of SIV reactivation after ATI, we first measured cell-associated (CA) viral gag DNA levels during cART treatment in macaques starting at 4, 10, or 27 dpi (*Figure 2a*). These data showed an early phase of rapid decline, followed by a slower decay with prolonged treatment. We then focused on the SIV gag DNA level in PBMC at the time of treatment interruption as a predictor of the frequency of reactivation. The levels of SIV gag DNA at interruption varied over a wide range between cohorts, ranging from <3.2 copies/$10^6$ PBMC among macaques treated beginning on day four post-infection, to as many as 1000 DNA copies/$10^6$ PBMC among animals treated beginning on day 10 (*Figure 2b*).

When we correlated the level of log SIV DNA at treatment interruption with the log frequency of reactivation, we found a significant linear correlation (linear regression slope = 0.20, $R^2$ = 0.56, p=0.0019)(*Figure 2b*). Although individual SIV genomes may vary greatly in their probability of reactivation (for example due to replication competence, integration site, or cell phenotype, activation state, and epigenetic or transcriptional blockades), if SIV DNA measured in the animals treated on different days had on average a similar probability of contributing to SIV reactivation from latency, then a doubling of DNA would equate to a doubling of reactivation (and we should expect a 1:1 correlation in this log:log plot). Instead, the slope was only 0.20. Although the day 4 and day 27 treated animals were treated for a similar length of time (*Figure 1b*), the day 27 treated animals had at least 116 fold higher SIV DNA level at interruption, but only an approximately 2-fold difference in the frequency of reactivation with a mean of 0.61 (SD 0.17, n = 6) versus 1.33 (SD 0.90,n = 5) reactivations per day for day 4 and day 27 treated animals respectively. To compare the 'per cell' frequency of reactivation between the groups, we estimated reactivations per DNA copy (*Figure 2c,d*). This analysis showed that treatment later in infection was associated with a declining frequency of reactivation per DNA copy measured at treatment interruption (*Figure 2c*). In addition, within the d4 treated and d27 treated animals, increased duration of treatment also showed a trend towards reduced per cell frequency of reactivation (*Figure 2d*). Thus, although SIV CA-DNA quantity in PBMC increases greatly from day 4 to day 27 post-infection, the frequency of reactivation per DNA copy declines in this time, apparently dependent on other factors.

## Contribution of cellular and viral activation

Differences in the per-cell probability of activating and producing progeny virus may explain the unexpected relationship between post-ATI reactivation frequency and SIV DNA copies in PBMC. Unspliced viral RNA transcription on cART has previously been used as a potential measure of the level of active transcription of infected cells (*Elliott et al., 2014*; *Petravic et al., 2017*), and some previous studies in HIV have suggested that levels of CA-RNA is a better predictor of time-to-recrudescence than CA-DNA (*Li et al., 2015*). Therefore, we also explored the relationship between the levels of unspliced SIV gag RNA throughout the course of treatment, including immediately prior to ATI, and the subsequent frequency of reactivation. A steep decline in CA-SIV-RNA is observed soon after treatment, which then slows at around day 100. This likely reflects the early, rapid loss of highly

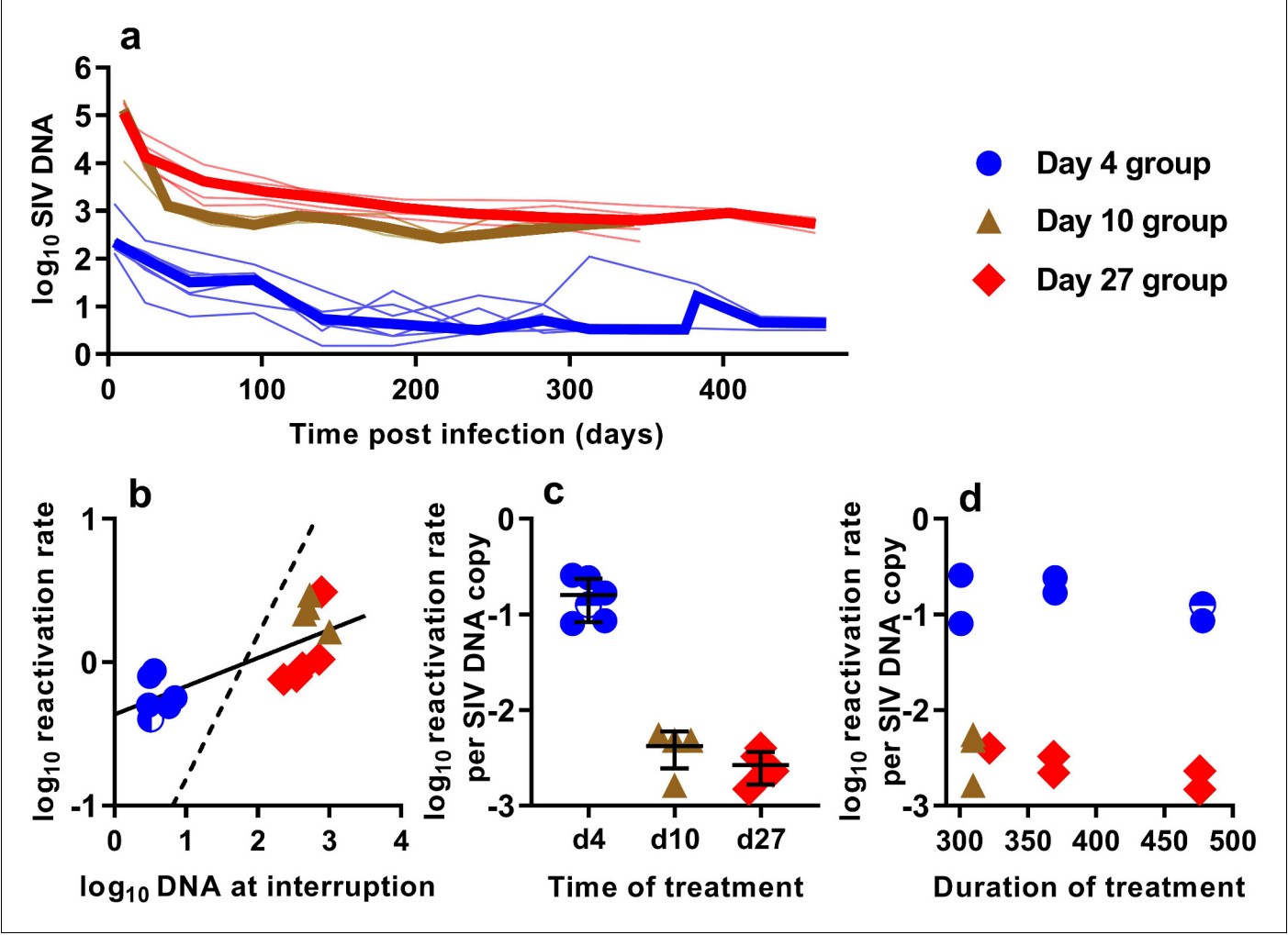

**Figure 2.** The relationship between SIV DNA and frequency of reactivation. (a) The levels of SIV CA-DNA in individual animals (thin lines) were measured in peripheral blood. Median is shown as thick line. (b) The relationship between frequency of reactivation from latency and SIV DNA levels at cART interruption, for individual animals treated at different times after infection. Linear regression line fitted to log-log transformed data, dashed line is the regression with the fixed slope = 1. (c) The frequency of reactivation per DNA copy for animals treated on different days post-infection, and (d) the relationship between duration of treatment and reactivation per DNA copy. Some animals treated at day four had undetectable DNA levels at interruption. In this case, we can only estimate an upper/lower bound and the circle is only shaded at the side of lower DNA values.

The online version of this article includes the following source data for figure 2:

**Source data 1.** Kinetics of cell-associated SIV DNA and its relationship to SIV reactivation frequency.

productive cells (with high RNA per DNA) due to viral cytopathic effects or immune clearance and subsequent survival of cells with a more restricted viral expression phenotype (low RNA per DNA) (*Figure 3a*).

Comparing reactivation rates with CA-RNA levels immediately prior to ATI, we find that CA-RNA is not a significant predictor of post-ATI reactivation frequency (Pearson r of log-log transformed data is 0.37, p=0.18, n = 15) (*Figure 3b*). Moreover, comparing between groups we again found that this was not a 1:1 relationship where a doubling of RNA would predict a doubling of reactivation. That is, the day 4-treated and d27-treated animals differed by 84 fold in their levels of SIV-CA RNA, despite only a two-fold difference in reactivation rate (*Figure 3b*). We also investigated the frequency of reactivation per RNA copy, and again found major differences between the groups (*Figure 3c*). As with the viral CA-DNA, CA-RNA is not proportionate to the frequency of post-ATI reactivation in vivo.

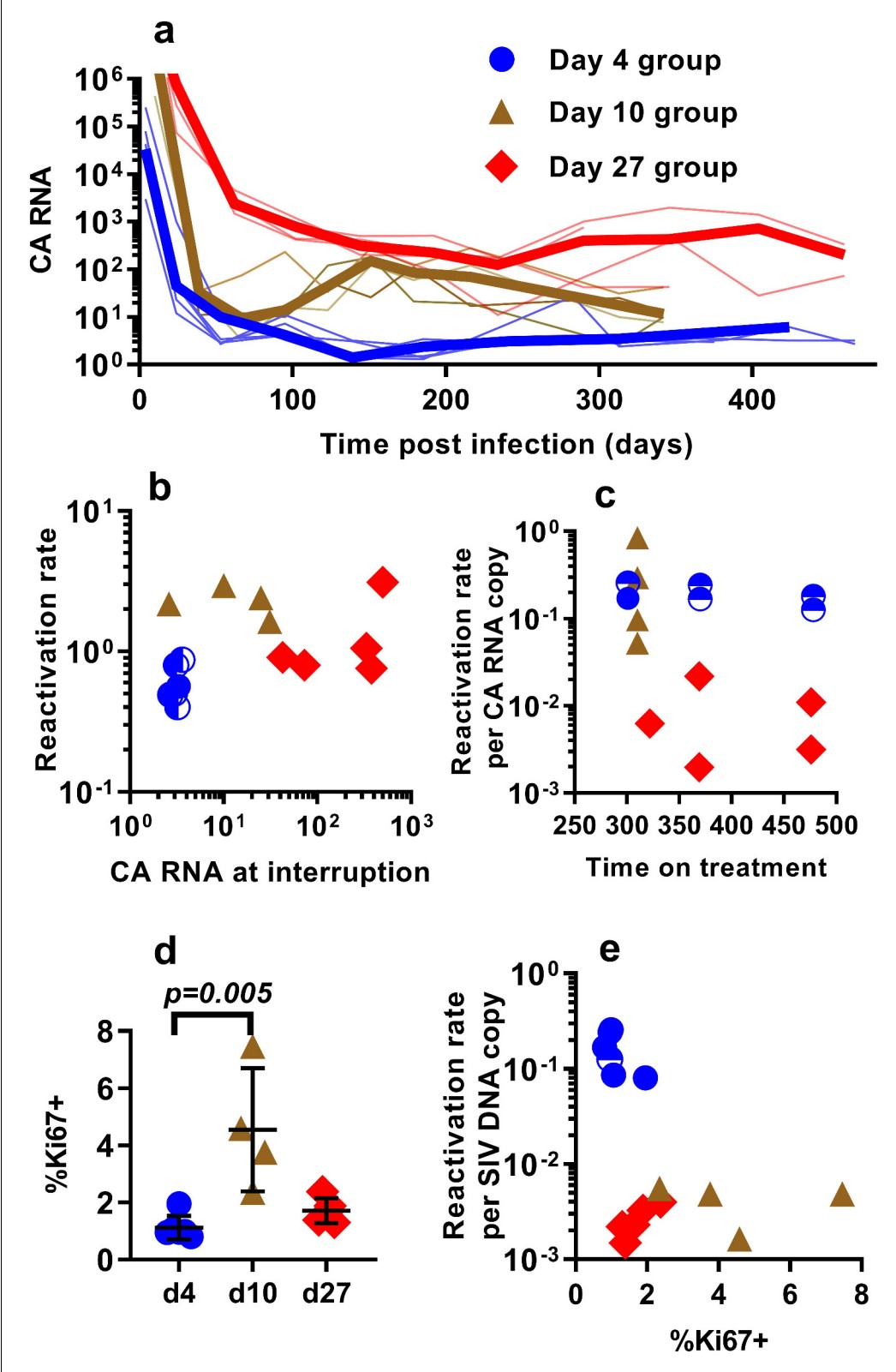

**Figure 3.** The relationship between CA-SIV-RNA, CD4[+] T cell activation, and frequency of reactivation. (a) The levels of CA-SIV RNA were measured over time in peripheral blood (thick line is the median). (b) The relationship between frequency of reactivation from latency and SIV RNA levels at interruption for individual animals treated at different times after infection. (c) The frequency of reactivation per RNA copy for animals treated on different days post-infection, according to the duration of treatment. (d) Immune activation was assessed based on expression of Ki67 in CD4[+] T cells in
*Figure 3 continued on next page*

*Figure 3 continued*
whole blood just prior to interruption. There was a trend to higher Ki67 expression in animals treated on day 10 and 27, and (**e**) reactivation per SIV DNA copy was negatively correlated with expression of Ki67 (Spearman r −0.56, p=0.034). Some animals treated at day four had undetectable RNA levels at interruption. In this case, we can only estimate an upper/lower bound and the circle is only shaded at the side of possible lower DNA values. Ki67 analysis was performed on fresh cells for animals in the day 4 and day 27 groups, and frozen cells for animals in the days 10 group.
The online version of this article includes the following source data for figure 3:

**Source data 1.** Levels of cell-associated RNA and Ki67 expression and their association with reactivation frequency.

One possibility to explain these data is that SIV CA-RNA is a poor measure of cellular activation leading to virus production and that other measures of immune activation may better predict the per cell SIV reactivation frequency. We therefore investigated other indicators of immune activation, studying CD4$^+$ and CD8$^+$ T cell expression of the activation markers CD38, HLA-DR, and Ki67 (these are generally correlated, so we limit our discussion to Ki67 expression in CD4$^+$ T cells). Comparing animals treated beginning at day 4, 10 and day 27, we found that the animals treated beginning at day 10 had higher immune activation just prior to treatment interruption than those treated on day 4 and 27 (day 4 - mean (+ /- SD) Ki67 = 1.12%, (+ /- 0.38%), n = 6; day 10 – mean = 4.54%, (+ /- 1.87%), n = 4; day 27 – mean = 1.72%, (+ /- 0.39%), n = 5; however, only day 4 and day 10 groups were significantly different, Dunn's multiple comparisons test's p=0.005) (*Figure 3d*). Overall, cellular activation in PBMC did not seem a major driver of reactivation frequency, since reactivation per SIV DNA copy was negatively correlated with the level of Ki67 expression (Spearman r −0.56, p=0.034).

## T cell responses to SIV peptides do not predict frequency of SIV reactivation

Our approach to estimating the average frequency of SIV reactivation from latency requires measuring the proportional contribution of different SIVmac239M clonotypes to the pool of total rebound plasma viremia after cART discontinuation. Thus, it estimates the frequency of successful reactivation events, and if some reactivating cells were targeted by immune responses, this would lead to an underestimation of the frequency of reactivation. Thus, it is possible that the observed differences in per cell reactivation frequency may be explained by increased immune control of a proportion of attempted reactivation events. To explore this hypothesis, we measured T cell responses (*Figure 4a*) using intracellular cytokine staining. We analysed IFN-γ, TNF-α, IL-2, CD107a and MIP1β responses to pooled peptides from SIV Env, Gag, Pol and accessory (ACC) proteins (see Materials and methods).

The proportion of CD8$^+$ T cells expressing CD107a in response to pooled SIV peptides (measured at the time of treatment interruption) was negatively correlated with the peak viral load observed during treatment interruption (Spearman r = −0.62, p=0.015, n = 15). This suggests that CD8$^+$ T cells are able to mediate some level of immune control of SIV viral growth during rebound (*Figure 4c*). We discuss here only the CD107a responses, while other responses are summarised in *Figure 4—figure supplement 1*, *2* and *3*.

Across the pooled animals from the day 4, day 10 and day 27-treatment initiation groups, there was a negative correlation between peripheral immune response and reactivation per DNA copy (Spearman r = −0.57, p=0.03, n = 15) (*Figure 4d*). However, this largely appeared driven by the differences between the groups; day four treated animals had lower immune responses, and they had higher reactivation per DNA than the day 27 treated.

However, if immune control directly reduced the frequency of reactivation, then we might also expect to see a similar trend within the groups, so that the animals with the highest immune response in each group would also have the lowest reactivation rate. Instead, we see that the animal in the day four group with the highest immune response also had the highest frequency of reactivation per DNA copy. Similarly, the trend within the day 27 treated animals is towards higher immune responses being associated with higher per-latent-cell reactivation. There is also one animal in each of the day 10 and d27 groups with immune responses as low as the d4 treated animals, but 60 and 100-fold lower reactivation per DNA copy. Although the small number of animals limits these comparisons, overall this suggests that mechanism of CD8$^+$ T cell immunity blocking or controlling a proportion of reactivation events is unlikely to explain the observed differences in per cell reactivation frequency.

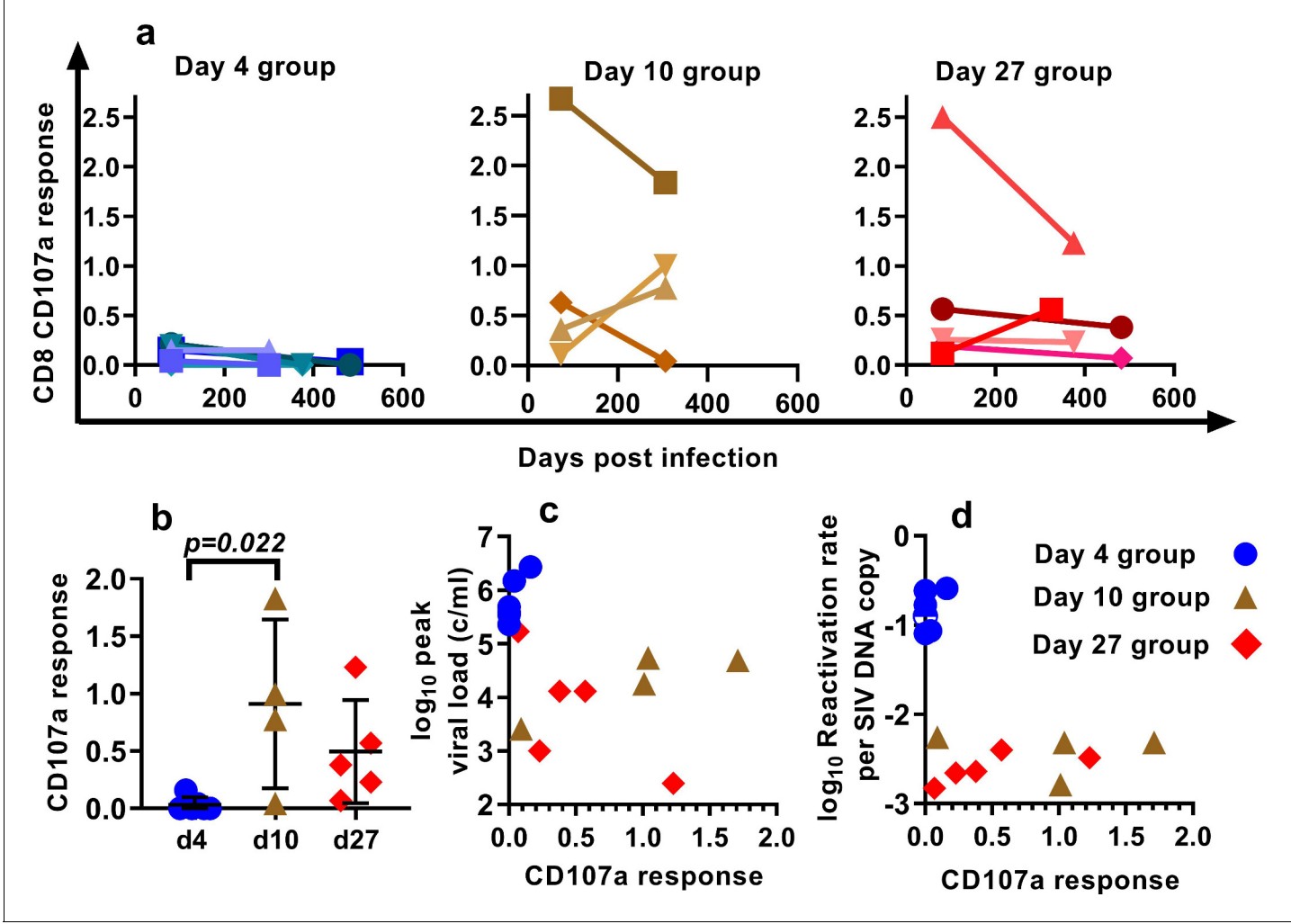

**Figure 4.** Effects of CD8[+] T cell response to SIV peptides. (a) The kinetics of the CD107a[+] response in CD8[+] T cells to pooled SIV peptides was measured in vitro using ICS (the sum of the responses to peptides from SIV ENV, GAG, POL and accessory (ACC) proteins is shown at the times indicated). (b) Animals treated beginning on day 10 and 27 have higher CD8[+] T cell responses to pooled peptides (%CD107a[+] cells shown) (Dunn's multiple comparisons test's p=0.022, for day 4 (n = 6) vs. day 10 (n = 4); p=0.053, for day 4 (n = 6) vs day 27 (n = 5)). (c) Relationship between peak viral load after treatment interruption and the level of CD107a response to pooled peptides (Spearman r = −0.62, p=0.015, n = 15). For other responses and peptides see *Figure 4—figure supplement 1*. (d) The dependence of reactivation rate per SIV DNA at time of interruption on the level CD107a response to pooled peptides, for other responses and peptides see *Figure 4—figure supplement 2*, for reactivation rate vs. immune response see *Figure 4—figure supplement 3*.

The online version of this article includes the following source data and figure supplement(s) for figure 4:

**Source data 1.** Relationship between immune response (CD8 cells expressing CD107a) and SIV reactivation.

**Figure supplement 1.** Effects of T cell response to SIV peptides on peak viremia after rebound.

**Figure supplement 1—source data 1.** Relationship between T cell response and peak SIV viremia after rebound.

**Figure supplement 2.** Effects of T cell response to SIV peptides on reactivation rate per SIV DNA copy.

**Figure supplement 2—source data 1.** Relationship between T cell response and SIV reactivation frequency per SIV DNA copy.

**Figure supplement 3.** Effects of T cell response to SIV peptides on reactivation rate.

**Figure supplement 3—source data 1.** Relationship between T cell response and SIV reactivation frequency.

## SIV DNA is largely replication competent

One potential explanation for our observed relationship between SIV DNA and reactivation frequency is that the proportion of SIV that is replication competent changes rapidly during early infection as a consequence of accumulation of deletions and or hypermutations. For example, if SIV DNA were 100% replication competent at day 4, and 1% replication competent at day 27, this could

explain a 100-fold change in reactivation rates. To investigate this, we sequenced near full length SIV DNA from PBMCs taken just prior to ATI from animals treated on day 10 (n = 4) and day 27 (n = 5)(we note that this data has also been included as part of another study; *Long et al., 2019*) (*Figure 5*). We found that 50/60 sequences from the day 10 treated and 86/101 sequences from day 27 treated animals were intact and presumptively replication competent (mean of 84% intact, range of 75–100% in individual animals). Since the majority of sequences were intact, this suggests that differences in reactivation frequency cannot be explained by differences in replication competence.

## A combination of SIV DNA copies and treatment duration is the best correlate/predictor of reactivation frequency

Thus far, we have investigated a variety of individual factors that might explain post-ATI reactivation frequency. In order to determine whether a combination of these factors could better predict reactivation frequency, we performed a multiple regression analysis that included all potential variables (listed in *Table 1*). We found that none of the peripheral immune or cytokine variables contributed significantly to the regression model, indicating that these variables cannot accurately predict reactivation frequency. The best model to predict reactivation frequency included both treatment duration and DNA levels at interruption (adjusted $R^2$ = 0.73).

## Modeling early establishment and rapid filling of the SIV reservoir in early treatment

Previous studies have demonstrated that latency is established very early after SIV infection (*Whitney et al., 2014*; *Whitney et al., 2018*; *Okoye et al., 2018*), consistent with our observations that seeding of the rebound competent viral reservoir is already established in macaques infected with a high-dose of virus and starting cART by day four post inoculation. However, our study suggests that the frequency of reactivation from latency does not increase much beyond early infection, even though large increases in SIV CA-DNA and CA-RNA are observed accumulating in PBMC. This raises the question of how the level of reactivation can saturate so early, while delayed treatment clearly allows much higher levels of infection and persistent SIV DNA? It seems likely that the majority of SIV reactivation events observed derive from a relatively small portion of the total

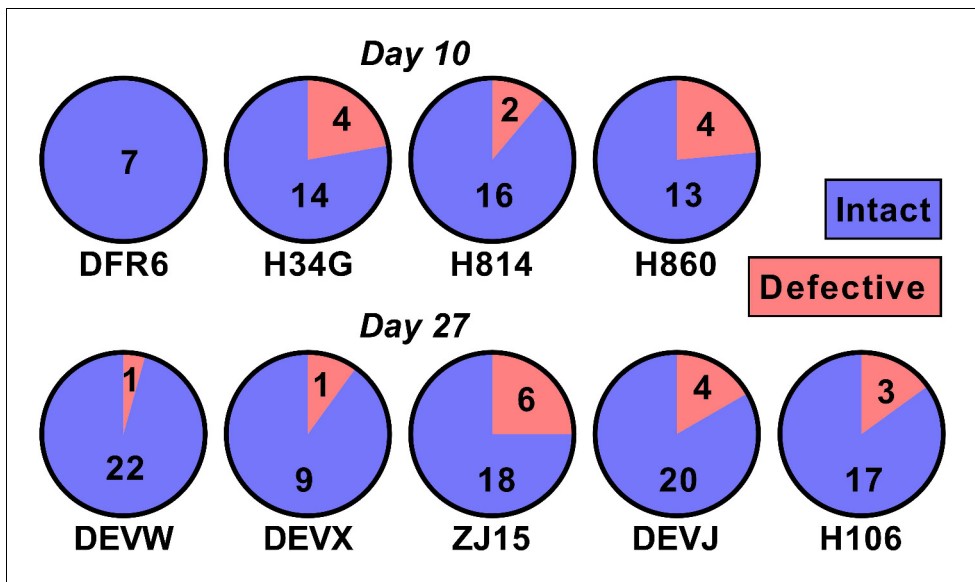

**Figure 5.** The majority of SIV DNA is intact. Full length SIV DNA was sequenced from PBMC of macaques just prior to treatment interruption. The number of intact versus defective sequences observed in each animal is indicated. The proportion of intact sequences was similar for animals treated on day 10 (top row) versus day 27 (bottom row). Overall 84% of SIV genomes were found to be intact.

The online version of this article includes the following source data for figure 5:

**Source data 1.** Proportion of intact and defective SIV DNA.

**Table 1.** Variables used in the multiple linear regression model.

| Variable number | Variable name | Allowed sign of coeffiecient |
| --- | --- | --- |
| 1 | Treatment Duration | -ve |
| 2 | $Log_{10}$(DNA at treatment) | +ve |
| 3 | $Log_{10}$(DNA at interruption) | +ve |
| 4 | $Log_{10}$(RNA at treatment) | +ve |
| 5 | $Log_{10}$(RNA at interruption) | +ve |
| 6 | $Log_{10}$(Viral Load at treatment) | -ve or +ve |
| 7 | $Log_{10}$(Area Under curve of Viral load at treatment) | -ve or +ve |
| 8 | $Log_{10}$(Maximum Viral Load) | -ve or +ve |
| 9 | CD107a | -ve |
| 10 | TNFα | -ve |
| 11 | IFNγ | -ve |
| 12 | IL2 | -ve |
| 13 | MIP1β | -ve |
| 14 | Ki67 | +ve |
| 15 | HLA-DR | +ve |
| 16 | CD38 | +ve |

proviral population, which can be established extremely early in infection. This 'reactivation initiating' population increases only slightly with ongoing viral replication, while infected cell frequency and CA-DNA levels accumulate, but does not proportionally contribute to post-ATI reactivation frequency. One additional possible explanation is that there may be limited and saturable cellular or anatomic compartments that are preferred for harbouring reactivatable viruses and include cells infected early after infection which persist during treatment. To investigate this, we adapted a standard model of HIV infection to investigate the effects of a 'susceptible subset' of cells on latency and reactivation. The model included a subclass of cells that was both more susceptible to infection (and thus infected earlier after inoculation), and more prone to later reactivation from latency (see Materials and methods). A subset of 0.4% of cells, that is 100-fold more susceptible to infection, and 500-fold more prone to reactivation (or a variety of combinations of these factors) could recapitulate the observed dynamics of DNA accumulation and post-ATI reactivation from the data (*Figure 6*).

## Discussion

A major question in HIV 'cure' research is whether biomarkers in peripheral blood correlate with and can be used to predict the duration of HIV remission after treatment interruption (*Margolis and Deeks, 2019*). We used variable timing of the initiation of cART, allowing different extents of viral dissemination before starting suppressive treatment, to produce SIV reservoirs of different sizes. The levels of SIV DNA at the time of ATI varied by >100 fold between different animals, from <3.2 copies/million PBMC in animals treated beginning on day 4, to >1000 copies in some animals starting treatment on day 10. The frequency of successful SIV reactivation after ATI also varied by ≈8-fold (from 0.4 to 3.10 reactivations per day). This is remarkable when one considers the different levels of exposure to infection. The peak viral load varied ≈350 fold between animals starting treatment on day 4 (geometric mean = $1.5 \times 10^5$ copies $ml^{-1}$, CI of geo. mean ($4.9 \times 10^4$, $4.4 \times 10^5$), n = 6) and animals treated on day 27 (geometric mean = $5.2 \times 10^7$ copies $ml^{-1}$, CI of geo. mean ($1.8 \times 10^7$, $1.5 \times 10^8$), n = 5). Similarly, the 'area under the curve' of viral exposure varied by >4,000 fold (geometric means $5.2 \times 10^4$, CI of geo. mean ($1.8 \times 10^4$, $1.5 \times 10^5$) n = 6, vs $2.2 \times 10^8$, CI of geo. mean ($1.3 \times 10^8$, $3.8 \times 10^8$), n = 5). However, although the prolonged viral replication led to an increase in SIV CA-DNA (*Archin et al., 2012*), the additional exposure to virus between day 4 and 27 was associated with only approximately double the frequency of reactivation. This suggests that most of the

additional infected cells, reflected in increased SIV CA-DNA generated between day 4 and day 27 did not significantly contribute to rebound viremia. The results suggest that key anatomic and/or cellular compartments that represent the major contributors to the rebound viremia may be limited and saturable as early as d4 after a high dose intravenous inoculation. The fact that very large changes in SIV CA-DNA are associated with only small changes in the frequency of reactivation perhaps goes some way to explaining why HIV DNA is a relatively poor predictor of time-to-reactivation after treatment interruption in many studies (*Li et al., 2015*; *Williams et al., 2014*; *Calin et al., 2016*; *Assoumou et al., 2015*).

One potential explanation for our observations that this SIV DNA that accumulates later in infection is not as replication competent as DNA integrated earlier in infection. Sequencing studies suggest that only 2.4% of HIV proviral DNA is intact in patients starting cART in chronic infection (*Hiener et al., 2017*; *Bruner et al., 2016*; *Bruner et al., 2019*). Thus, for example, if SIV proviral DNA integrated at day 4 was 100% replication competent, but DNA integrated after this were 99% defective, this might explain the observed relationship between reactivation frequency and SIV DNA copies. However, near full length viral sequencing from animals treated on day 10 and day 27 revealed that the vast majority (>80%) of sequences were intact. Studies of $SIV_{mac251}$ DNA in animals treated in chronic infection (95 weeks post-infection) for 36 weeks revealed that 28% remained intact (*Bender et al., 2019*). Thus, it appears that at least over the first year or two of infection, SIV DNA remains intact at much higher rates than HIV. Taken together, it is clear that the observed differences in reactivation frequency between animals treated on different days cannot be explained by different proportions of replication competent virus.

Previous work has suggested that extremely early treatment can either delay or prevent post-ATI SIV viral recrudescence (*Whitney et al., 2014*; *Whitney et al., 2018*; *Okoye et al., 2018*). In a study by Whitney et al, intrarectal inoculation of 500 TCID50 of SIVmac251 and 6 months of cART begun

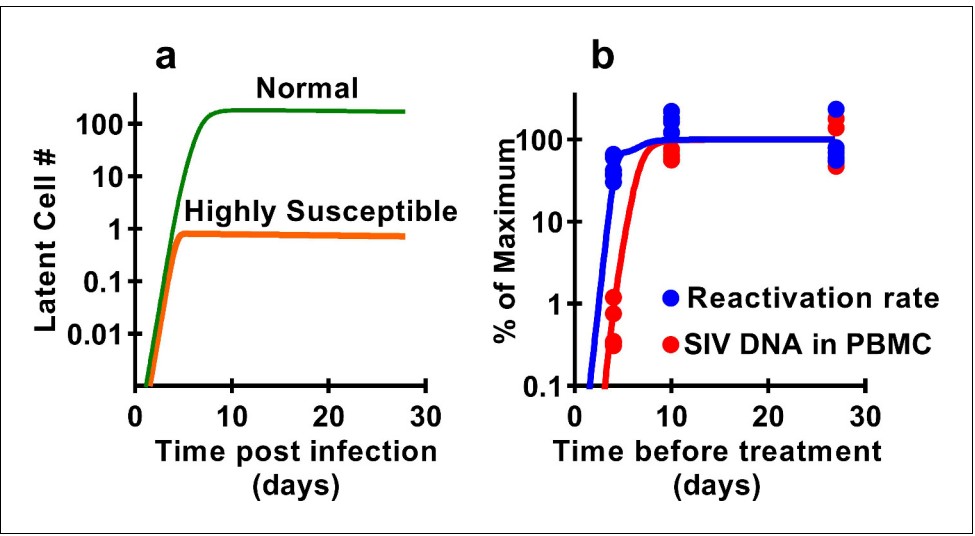

**Figure 6.** Modelling a subset of susceptible cells that are the major drivers of SIV reactivation. The frequency of reactivation from latency has reached about half of its maximum in animals treated 4 days post-infection, whereas the level of SIV DNA continues to increase around 100-fold between day 4 and day 10 post-infection. We use a modelling approach to explore the dynamics of early infection and consider the possibility that there are different subsets of infected cells. For example, if we had a subset of cells which is both highly susceptible to infection, and has a high frequency of reactivation from latency, then these cells would both be infected first (orange curve in panel a), and would also make an out-sized contribution to the overall frequency of reactivation from latency. The less susceptible and quiescent pool (green curve in panel a) continues to 'fill' over time, but this has relatively little impact on later reactivation rates The frequency of reactivation and SIV DNA levels include contributions from both subsets. However, the small subset of highly active cells can make a major contribution to the frequency of reactivation, while having little effect on total DNA levels. For example, panel **b** shows experimental data (dots) and along with panel a illustrates that a theoretical subset of only 0.4% of cells that is 100-fold more susceptible to infection, and 500-fold more prone to reactivation, could explain the data on SIV DNA and SIV reactivation frequency.

on day three post-infection (when animals did not have detectable plasma virus) resulted in delayed viral rebound. However, animals treated for 6 months beginning on days 7, 10 and 14 post infection showed similar rebound times (*Whitney et al., 2014*). *Okoye et al. (2018)* infected macaques with 500 IU of SIVmac239, and found that animals treated for nearly 2 years beginning on day 4–5 post infection (with viral loads of 60–1100 copies/ml at treatment initiation) failed to rebound after treatment interruption (*Okoye et al., 2018*). By contrast, our study involved intravenous inoculation of $2.2 \times 10^5$ IU of SIVmac239M viral stock, and animals treated on day four had maximal viral loads of between $3.8 \times 10^4$ and $9.1 \times 10^5$ (geometric mean of $1.5 \times 10^5$) copies/ml of plasma virus. Taken together, these data suggests that extremely early treatment (with viral loads at treatment <1000–10,000 copies/ml) may severely restrict the establishment of the RCR. However, once plasma viremia passes around 10,000 copies/ml the subsequent post-ATI frequency of reactivation is established. Extending the duration of infection prior to starting cART (from day 4 to day 27) may greatly increase the level of proviral DNA (*Archin et al., 2012*) and cell-associated RNA in PBMC, but does comparatively little to increase the frequency of reactivation.

Although animal models of infection provide the ability to control factors such as time and duration of treatment, it is not clear whether these same lessons are applicable to HIV infection. The RCR of SIV may differ from that in HIV, either because of the cells targeted for infection, or because of differences in infected cell behaviour. For example, cellular entry using alternative receptors such as CXCR6 and GPR15, or differences in host restriction factors may lead to differences in infected cell phenotypes in SIV (*Riddick et al., 2016*; *Reynolds et al., 2011*). Similarly, although coinfection of cells with multiple copies of HIV is thought to be low (*Josefsson et al., 2011*), rates of SIV coinfection of cells were not determined in our study and could play a role in determining the frequency of reactivation per DNA copy observed. However, recent studies have suggested that integrations sites and the propensity for clonal expansion of latent virus are similar between SIV and HIV (*Ferris et al., 2019*). The initial inoculum of virus in our studies is several orders of magnitude higher than that estimated in HIV (*Keele et al., 2008*), and thus this may disproportionately contribute to the early establishment and potential saturation of the SIV RCR. However, since viral loads increase around 10-fold per day in early infection (*Figure 1a,c,e*), it is unlikely that this initial inoculum virus equates to >0.1% of virus present even at day four post-infection (the timing of earliest treatment). Thus, it seems unlikely that the higher inoculum in SIV infection could be a major contributor to the SIV reservoir measured later during treatment.

Our finding of early 'filling' of the RCR appear consistent with the more limited observations of reactivation in early HIV infection. In particular, although some studies have shown delayed post-ATI rebound in HIV patients treated early in infection and with small reservoirs (*Li et al., 2015*; *Steingrover et al., 2008*), the size of the reservoir appears only weakly associated with the time to recrudescence of HIV after treatment interruption (*Li et al., 2015*; *Williams et al., 2014*). A number of recent studies have shown that even in individuals treated extremely early and/or with a very small reservoir, the time to viral recrudescence can be very similar to that observed in individuals treated beginning in chronic infection and with a presumably larger reservoir (*Calin et al., 2016*; *Assoumou et al., 2015*). So while the case study of the Mississippi baby suggests that extremely early treatment can significantly delay viral rebound (*Luzuriaga et al., 2015*), Colby et al have recently shown that individuals treated very early during Fiebig stage one have only slightly longer average time to detection of plasma viral rebound than individuals starting treatment in chronic infection (median 26 vs. 14 days), despite considerably lower HIV DNA levels (*Colby et al., 2018*). This strongly supports our observations with SIV, in that the frequency of post-ATI viral reactivation is not only established extremely early in infection, but once established does not increase much from early primary infection to chronic infection.

A significant limitation of our study is that we were not able to perform quantitative viral outgrowth assays (QVOA) (*Finzi et al., 1999*) on our macaque samples, due to limitations in available cell numbers in rhesus macaques. Comparison of HIV sequences between circulating provirus and ex vivo-inducible virus show that there is a significant sequence overlap between these two compartments (*Lu et al., 2019*). However, a recent study showed that despite the overlap between provirus and QVOA, there was no overlap of either compartment with post-ATI rebound virus (*Lu et al., 2019*). This was taken to suggest that rebound virus originates from a different pool of cells to that reflected in circulating provirus. Combined with our SIV results, it is clear that most current measures of the HIV reservoir are in fact measuring total CA-HIV or even presumptively intact proviruses from

PBMCs (*Bullen et al., 2014*) but not the major contributor to post-ATI rebound viremia. Further work is clearly required to focus attention on what must be a small proportion of the total latently infected cells (compared to total provirus in PBMC), that appear to be the major drivers of rebound.

The observation that DNA levels do not predict reactivation frequency between individuals does not answer the question of whether it might be a useful predictor of the impact of anti-latency therapies within an individual. Thus, for example, if the RCR declines (or is reduced by treatment) in proportion to the total HIV DNA, then monitoring total DNA might still be useful to assess the impact of anti-latency treatments on reactivation frequency within an individual patient over time. The stability of the HIV reservoir has been measured in humans by measuring the decay of both HIV DNA and ex vivo inducible HIV, which were both found to decay with a half-life of >4 years (*Besson et al., 2014*; *Siliciano et al., 2003*). Our study suggests that, after 6 months of therapy, SIV DNA decays with a half-life of around 345 days and SIV CA-RNA decays faster (half-life = 86 days, p<0.016, LME model). The frequency of reactivation declines slightly faster than SIV-DNA with a half-life of $\approx 216$ days (although not significantly different, p=0.53, LME model). Thus, reactivation frequency appears to decay at the same or faster rate than DNA levels in PBMC.

It is important to understand the mechanisms of latency formation, maintenance and reactivation when designing interventions aimed to induce long-term remission. Our work suggests that the RCR can be established very early following infection and provides some explanation as to why treating early may only have a small effect on subsequent time-to-recrudescence of virus (even if total HIV DNA levels continue to increase). A major impediment in HIV remission studies is finding a proxy measure that can predict the time-to-recrudescence of virus, without patients undergoing the risks of treatment interruption (*Julg et al., 2019*). This may be challenging, as our studies suggest that total PBMC SIV CA-DNA is not the main determinant of post-treatment rebound. Our work also suggests that efforts to develop optimal interventions to prevent HIV reactivation following treatment interruption will require a more thorough understanding of viral latency.

## Materials and methods

### Animals

Nine purpose-bred Indian-origin male rhesus macaques (*Macaca mulatta*) were housed at the National Institutes of Health (NIH) and cared for in accordance with the Association for the Assessment and Accreditation of Laboratory Animal Care (AAALAC) standards in an AAALAC-accredited facility and all procedures were performed according to protocols approved by the Institutional Animal Care and Use Committee of the National Cancer Institute (Assurance #A4149-01). Animal care was provided in accordance with the procedures outlined in U.S. NIH *Guide for Care and Use of Laboratory Animals*. Reference numbers associated with the ethical approval are AVP047 and AVP058.

At the start of the study, all animals were free of cercopithecine herpesvirus 1, simian immunodeficiency virus (SIV), simian type-D retrovirus, and simian T-lymphotropic virus type 1.

### Infection and treatment

In total, 15 animals were intravenously infected with of transfection produced SIVmac239M as previously described (*Fennessey et al., 2017*). 9 of 15 animals were infected with $2.2 \times 10^5$ IU (1 mL) and treated with cART at 4 and 27dpi, while the remaining four animals were infected with $1 \times 10^4$ IU (1 mL) and treated with cART at 10dpi. cART regimen for d27 and d4 animals consisted of a co-formulated preparation containing the reverse transcriptase inhibitors tenofovir (TFV: (R)−9-(2-phosphonylmethoxypropyl) adenine (PMPA), 20 mg/kg) and emtricitabine (FTC; 50 mg/kg) administered by once-daily subcutaneous injection, plus raltegravir (RAL; 150–200 mg) given orally twice daily. Day four animal were additionally treated with the protease inhibitor indinavir (IDV; 120 mg BID) and ritonavir (RTV; 100 mg BID) for the first 9 months (*Fennessey et al., 2017*). Day 10 animals received a co-formulated preparation of emtricitabine (FTC, 40 mg/kg), tenofovir disoproxil fumarate (TDF, 5.1 mg/kg), and dolutegravir (DTG, 2.5 mg/kg) administered by once-daily subcutaneous injection.

### Barcode sequencing and enumeration

Sequence analysis was used to enumerate the number of detectable barcodes measured during primary infection prior to cART and during rebound after cART interruption. Viral nucleic acid was sequenced using next generation sequencing as previously described (*Fennessey et al., 2017*).

### Plasma viral load determination

Plasma viral load determinations for SIV RNA were performed over the duration of the study using quantitative real-time PCR as described previously (*Li et al., 2016*). The limit of detection of this assay is 15 vRNA copies/mL.

### Quantitative evaluation of cell-associated DNA and RNA

Quantitative assessment of cell-associated viral DNA and RNA in PBMC pellets was determined by the hybrid real-time/digital RT-PCR and PCR assays essentially as described previously (*Hansen et al., 2011*) but specifically modified to accommodate cell pellets. 100 µL of TriReagent (Molecular Research Center, Inc) was added to cell pellets in standard 1.7 mL microcentrifuge tubes and the tubes sonicated in a Branson cup horn sonicator (Emerson Electric, St. Louis) for 15 s at 60% amplitude to disrupt the pellet. Additional TriReagent was added to a final volume of 1 mL and the remainder of the protocol was carried out as described previously (*Hansen et al., 2011*). Limit of detection is evaluated on a sample by sample basis, dependent on the number of diploid genome equivalents of extracted DNA assayed.

### Flow cytometry

Antibodies and reagents were obtained from BD Biosciences, unless indicated otherwise, and data analysis was performed using FCS Express (De Novo Software). Antibody panel validation and population gating were performed using fluorescence-minus-one and corresponding biological controls. Compensation was performed using goat-anti-mouse Ig (Spherotech), anti-Rat Ig and amine-reactive (Invitrogen) compensation bead, single-color controls, under identical sample treatment conditions. For activation immunophenotyping, 100 µl EDTA-anti-coagulated whole blood or $1 \times 10^6$ quick-thawed cryopreserved PBMC were incubated with the following antibody panel: CD4 Pacific Blue (OKT4, BioLegend), CD8 BV510 (SK1), CD14 BV605 (M5E2, BioLegend), CD69 BV650 (FN50), CD163 BV711 (GHI/61), CCR5 PerCP-Cy5.5 (3A9), CD38 PE (OKT10; NIH Nonhuman Primate Reagent Resource), CD28 ECD (CD28.2, Beckman Coulter), CD95 PE-Cy5 (DX2), HLA-DR Alexa Fluor 700 (L243, BioLegend) and CD3 APC-Cy7 (SP34-2). Samples were lysed with 1X BD FACS Lyse buffer, washed and then fixed and permeabilized with BD Cytofix/Cytoperm reagents, according to manufacturer instructions. Samples were incubated with an intracellular staining panel containing Ki67 BV786 (B56), MNDA FITC (3C1, Beckman Coulter), CD68 PE-Cy7 (Y1/82A, BioLegend) and CD66 APC (TET2, Miltenyi Biotech), washed and approximately 200,000 CD3$^+$ T-cells were acquired for each sample using a BD LSR-II flow cytometer. For intracellular cytokine staining, $1 \times 10^6$ quick-thawed, DNase I-treated (30 U/ml, Roche) cryopreserved PBMC were stimulated for 8 hr in 96-well polypropylene round-bottom plates (Costar) with pools of 2 µg/ml overlapping 15mer SIVmac239 accessory (combination of nef, rev, tat, vif, vpr and vpx), gag, pol and env peptides (NIH AIDS Reagent Program), in the presence of CD107a BV785 antibody (H4A3, BioLegend). Phorbol 12-myristate 13-acetate with Ionomycin (Sigma) and DMSO in media were used as positive and negative controls, respectively, and 5 µg/ml brefeldin A (Sigma) mixed with 0.14 µl/well BD GolgiStop was added after 1 hr of stimulation to block protein transport. Samples were cultured in a DigiTherm Unibator (Tritech Research) at 37°C, 5% $CO_2$ and immediately after the 8 hr incubation, rapidly cooled and maintained at 4°C until the following morning. Samples were surface stained with an antibody panel containing Yellow Fluorescent Reactive Dye (Invitrogen), CD4 Pacific Blue, CD28 ECD, CD95 PE-Cy5 and CD8α PE-Cy7, fixed and permeabilized with BD Cytofix/Cytoperm reagents and then incubated with an intracellular staining panel containing CD3 APC-Cy7, IFNγ FITC (B27), MIP-1β PE (D21-1351), TNF-α BV711 (MAb11, BioLegend) and IL-2 APC (MQ1-17H12). Cells were washed and approximately 200,000 live CD3$^+$ T-cells were acquired for each sample using an HTS-equipped BD LSR-II cytometer.

## Single genome amplification and sequencing of full-length SIV genome

Genomic DNA was purified from PBMC from SIV-infected rhesus macaques using QIAamp DNA Mini Kit (Qiagen). SGA sequencing was performed by diluting template DNA such that the majority of wells contain no template and the wells with template most likely contain only a single copy (*Keele et al., 2008*). Nested PCR was performed using the following primers: SIVnFL.F1 5′-GAT TGG CGC CYG AAC AGG GAC TTG-3′; SIVnFL.R1 5′-CCC AAA GCA GAA AGG GTC CTA ACG-3′ for the first round PCR and SIVnFL.F2 5′-GTG AAG GCA GTA AGG GCG GCA GG-3′; SIVnFL.R2 5′-CCA GGC GGC GRC TAG GAG AGA TGG-3′ for the second round. The PCR reaction consisted of 1x SuperFi buffer (Invitrogen), 0.2 mM each of dNTPs, 1x SuperFi GC enhancer (Invitrogen), 0.02 U/uL Platinum SuperFi DNA Polymerase (Invitrogen), 0.5 uM forward primer, 0.5 uM reverse primer, and template DNA. Thermal cycling conditions were as follows: 95℃ for 2 min; 35x (95℃ for 10 s, 68℃ (round 1) or 72℃ (round 2) for 10 s, 68℃ (round 1) or 72℃ (round 2) for 5 min; and a final extension of 5 min followed by 4℃ hold. The higher annealing and extension temperatures (72℃) in round two were used to avoid mispriming. SGA PCR products were directly sequenced using BigDye Terminator Sanger Sequencing (Life Technologies) with the primers previously described (*Lopker et al., 2016*).

## Estimating the reactivation rate

In order to estimate the reactivation rate, we used a method that incorporates the ratio of the number of copies of different barcoded clonotypes, and the growth rate of virus (*Fennessey et al., 2017*). We assumed that all barcodes have approximately the same growth rate and, thus, the reactivation rate (RR) can be estimated using

$$RR = \frac{g(n-1)}{\sum_{i=1}^{n-1}(lnS_{i+1} - lnS_i)},$$ (1)

where $g$ is the estimate of the growth rate of the single barcode, $S_i i = 1,..,n$ is the number of sequences for each barcode. The schematic explanation of this method is presented in the *Figure 1d*.

The ratio of number of copies of each clonotype was estimated from Illumina sequencing of plasma virus, and growth rate was estimated as the maximal growth rate between any two viral load measurements in each animal. We assumed the exponential growth of virus between two neighbouring measurement as shown below

$$V(t_{i+1}) = V(t_i)e^{g_i(t_{i+1} - t_i)}, \ i = 1,...,m-1,$$ (2)

where $m$ is the number of measurements of viral load, $V(t_i)$ is the viral load at time $t_i$. Thus the estimate of the growth rate on the interval can be found using formula (*Deeks et al., 2016*)

$$g_i = \frac{lnV(t_{i+1}) - lnV(t_i)}{t_{i+1} - t_i}, i = 1,..,m-1,$$ (3)

and the maximal viral load in subject is:

$$g = \max_{i=1,..,m-1} g_i$$ (4)

Note that the maximal two-point growth gives slightly higher growth rates than fitting of multiple timepoints, but allowed for consistency across all groups, to avoid issues related to different sampling times.

## Estimating the area under the curve of viral load

In order to estimate the area under the curve we approximated the trajectory of viral load by a piecewise function (*Fennessey et al., 2017*) where growth rates on the interval can be found using formula (3). Having this simple approximation, we can find the integral of this function between any timepoints. This was implemented in Wolfram Mathematica 11.2, Wolfram Research Inc, Champaign, IL, USA.

## Statistical analysis

All statistical tests were performed in GraphPad Prism 7.04, GraphPad Software, La Jolla, CA, USA.

## Estimating the decay of SIV DNA, CA RNA and reactivation rate

In order to estimate the decay rate of CA RNA and SIV DNA in chronic infection, we estimated decay rates after approximately 6 months of treatment. We also measured the frequency of reactivation (using the Materials and method described above) in monkeys treated for different times. We fitted the following linear mixed effect model (*Eriksson et al., 2013*) to ln-transformed data.

$$y = A_i + a_{ij} + (B_i + b_{ij})x_{ij} + \varepsilon_{ij},\qquad(5)$$

where $A_i$ and $B_i$ are fixed effect intercept and slopes where index $i$ corresponds to different types of data such as SIV CA-RNA or CA-DNA. Index $j$ in random effect parameters *aij* and *bij* corresponds to different monkeys in case of CA RNA and SIV DNA or in case of reactivation rate to groups of monkeys treated on day 4 or day 27. This model was implemented in R (v. 3.3.1, The R Foundation for Statistical computing) using standard function *nmle* for fitting and performing statistical tests.

## Multiple regression methods

Multiple linear regression was performed using both forwards stepwise regression. Criteria for inclusion in the model was that the variable had a significance value (p-value) of greater than 0.05, and that the regression coefficient was in a biologically meaningful direction (*Table 1*).

## Modelling infection and 'filling' of a susceptible reservoir

The model considers two populations of cells. One group is comprised of normal cells, and the other contains highly susceptible cells that are also highly prone to post-treatment reactivation. When target cells become infected, some fraction of them, *f*, become productively infected cells. Then (*1-f*) of them become latently infected cells. Highly susceptible cells are infected with a *k* fold higher infectivity rate than normal cells. We assume that reactivation rate is the sum of reactivation from both the normal cells and the highly reactivatable cells, but the highly reactivatable cells contribute *a* times as much to the reactivation rate.

The equations are:

$$\frac{dT_N}{dt} = -\beta V T_N$$

$$\frac{dT_S}{dt} = -k\beta V T_S$$

$$\frac{dI}{dt} = f\beta V(T_N + kT_S) - \delta_I I$$

$$\frac{dL_N}{dt} = (1-f)\beta V T_N - \delta_L L_N$$

$$\frac{dL_S}{dt} = (1-f)k\beta V T_S - \delta_L L_S$$

$$\frac{dV}{dt} = pI - cV$$

Where $T_N$ and $T_S$ are normal and susceptible target cells respectively, $I$ are infected cells, $L_N$ and $L_S$ are normal and susceptible latent cells respectively and $V$ is free virus. The parameters of the model are given by: $\beta$, the infection rate of normal targets, $k$ the increase in infection rate for susceptible targets, $f$, the fraction of target cells that become productively infected (not latent), $\delta_I$, the death rate of productively infected cells, $\delta_L$, the death rate of latent cells, $p$, the rate of production of virus and $c$, the death rate of virus.

The reactivation rate is determined as:

$$RR \propto L_N + aL_S$$

Parameters of the model.

$T_S(0) = 0.004 T_N(0)$ (i.e. Number of highly susceptible cells is initially 0.4% of the number of normal cells),

$V_0 = 40$, $f = 0.2$ (i.e. 80% of cells become infected and 20% become latent), $\beta = 1.4 \times 10^{-7}$, $k = 100$, $a = 500$.

$T_N(0) = 1000$ cells, $\delta_I = 1/\text{day}$, $\delta_L = \frac{1}{200}/\text{day}$, $c = 20/\text{day}$, $p = 5 \times 10^5/\text{day}$.

# Acknowledgements

The following reagents were obtained through the AIDS Reagent Program, Division of AIDS, NIAID, NIH: SIVmac239 Nef, Rev, Tat, Vif, Vpr, Vpx, Gag, Pol and Env Peptide Sets. We thank the nonhuman primate care staff in the Laboratory Animal Sciences Program, Frederick National Laboratory for Cancer Research for expert animal care. We also thank the Nonhuman Primate Research Support and the Quantitative Molecular Diagnostics Cores of the AIDS and Cancer Virus Program, Frederick National Laboratory for Cancer Research for specimen processing, animal support and viral load analysis. This project has been funded in whole or in part with Federal funds from the National Cancer Institute, National Institutes of Health, under Contract No. HHSN261200800001E. The content of this publication does not necessarily reflect the views or policies of the Department of Health and Human Services, nor does mention of trade names, commercial products, or organizations imply endorsement by the U.S. Government.

# Additional information

## Funding

| Funder | Grant reference number | Author |
| --- | --- | --- |
| National Institutes of Health | HHSN261200800001E | Christine M Fennessey<br>Carolyn Reid<br>Charles M Trubey<br>Jeffrey D Lifson<br>Brandon F Keele |
| National Health and Medical Research Council | 1052979 | Mykola Pinkevych<br>Deborah Cromer<br>Miles P Davenport |
| National Health and Medical Research Council | 1149990 | Mykola Pinkevych<br>Deborah Cromer<br>Miles P Davenport |
| National Health and Medical Research Council | 1080001 | Miles P Davenport |

The funders had no role in study design, data collection and interpretation, or the decision to submit the work for publication.

## Author contributions

Mykola Pinkevych, Resources, Data curation, Software, Formal analysis, Validation, Investigation, Visualization, Writing—original draft, Writing—review and editing; Christine M Fennessey, Carolyn Reid, Charles M Trubey, Data curation, Validation, Investigation, Writing—original draft, Writing—review and editing; Deborah Cromer, Software, Formal analysis, Investigation, Visualization, Writing—original draft, Writing—review and editing; Jeffrey D Lifson, Conceptualization, Supervision, Methodology, Writing—original draft, Project administration, Writing—review and editing; Brandon F Keele, Conceptualization, Supervision, Funding acquisition, Validation, Investigation, Methodology, Writing—original draft, Project administration, Writing—review and editing; Miles P Davenport, Conceptualization, Formal analysis, Supervision, Funding acquisition, Validation, Investigation, Methodology, Writing—original draft, Project administration, Writing—review and editing

## Author ORCIDs
Miles P Davenport ⬦ https://orcid.org/0000-0002-4751-1831

## Ethics

Animal experimentation: Animals were cared for in accordance with the Association for the Assessment and Accreditation of Laboratory Animal Care (AAALAC) standards in an AAALAC-accredited facility and all procedures were performed according to protocols approved by the Institutional Animal Care and Use Committee of the National Cancer Institute (Assurance #A4149-01). Animal care was provided in accordance with the procedures outlined in the "Guide for Care and Use of Laboratory Animals". Reference numbers associated with the ethical approval are AVP047 and AVP058.

## Decision letter and Author response

Decision letter https://doi.org/10.7554/eLife.49022.sa1
Author response https://doi.org/10.7554/eLife.49022.sa2

## Additional files

### Supplementary files

• Transparent reporting form

### Data availability

Source data files have been provided for the figures.

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
