## [Decision Letter]

**Acceptance summary:**

HIV eradication is the ultimate goal of AIDS research and some strategies are evaluated in clinical trials. One caveat of these studies is that currently no validated biomarkers that accurately predict the efficacy of these strategies in prolonging virus-free remission after treatment interruption are available. The present study addresses this important issue by evaluating various immune and virologic biomarkers as potential correlates of time to viral rebound during treatment interruption using barcoded SIVmac239 constructs in the macaque model. The study shows that very early initiation of treatment (4 dpi) reduces the frequency of reactivation, while initiation at 10 or 27 dpi hardly made a difference. This very early treatment strongly reduced the levels of proviral DNA; surprisingly, however, a 100-fold difference in the proviral copy numbers in blood PBMCs was only associated with a 2-fold difference in reactivation frequencies from latency. Thus, the data indicate that peripheral blood measures of the reservoir are insufficient to predict reactivation from latency. Although the results are largely negative they are clearly of significant interest for everybody interested in the cure of HIV/AIDS.

**Decision letter after peer review:**

Thank you for submitting your article "SIV reactivation from latency is established early and not predicted by standard measures of reservoir" for consideration by *eLife*. Your article has been reviewed by 3 peer reviewers, including Frank Kirchhoff as the Reviewing Editor and Reviewer #1, and the evaluation has been overseen by a Reviewing Editor and Neil Ferguson as the Senior Editor. The following individuals involved in review of your submission have agreed to reveal their identity: Adam Mitchell Spivak (Reviewer #2).

The reviewers have discussed the reviews with one another and the Reviewing Editor has drafted this decision to help you prepare a revised submission.

Summary:

Although HIV eradication is the ultimate goal of AIDS research and some strategies are currently evaluated in clinical trials, there are no validated biomarkers that accurately predict the efficacy of these strategies in prolonging virus-free remission after treatment interruption. Davenport and colleagues address this important issue by evaluating various immune and virologic biomarkers as potential correlates of time to viral rebound during treatment interruption using barcoded SIVmac239 constructs in the macaque model. Using a previously established mathematical model, the authors estimate the number of reactivation events in each animal after ART cessation that contributed to rebound viremia. These rebound rates were compared to measures of SIV DNA and SIV CA-RNA as well as CD8 responses to in vitro SIV peptide pulsing experiments. The authors show that very early treatment initiation (4 dpi) reduces the frequency of reactivation, while initiation at 10 or 27 dpi hardly made a difference. This very early treatment reduced the levels of proviral DNA by about two orders of magnitude but was only associated with a 2-fold difference in reactivation frequencies from latency. Altogether, no stringent relationship between the modelled SIV reactivation rate and any of the virologic and immunologic parameters investigated could be established. Thus, the authors conclude that peripheral blood measures of the reservoir are insufficient to predict reactivation from latency.

All reviewers agree that the study is well performed and addresses a very important and timely question. Limitations are that findings are largely negative and that (as discussed by the authors) no viral outgrowth assays could be performed since available cell numbers are limited in the SIV/macaque model. While the results are clearly of significant interest several concerns should be addressed.

Essential revisions:

1) The manuscript focuses almost exclusively on the measurement of total proviral DNA, and does not directly take into account that in humans this measurement vastly over-estimates the reservoir size due to the accumulation of defective proviruses that functionally may not contribute to the reservoir at all. In the discussion of this work, the authors dismiss this phenomenon by pointing to an upcoming manuscript evaluating this in the SIV model and provide a very high frequency of intact proviruses (80%, section). No data or methods accompany this, and it is markedly different than what is seen in humans (2% of total proviruses are thought to be genetically intact) – which raises some concerns about the nature of this marked difference and the applicability of this model to the human HIV reservoir. This should be addressed in more detail or (better) substantiated by experimental data.

2) The viral inoculum that gives rise to primary infection here is many orders of magnitude larger than what is thought to cause primary infection in humans (at least via sexual transmission), and this may affect the kinetics of reservoir establishment – making this a less-than-ideal model of human reservoirs. How much of the concept of 'early / rapid filling' of the reservoir shown here is an artefact of the non-human primate SIV infection model, in which many thousands of intact virions are used simultaneously for primary infection? This quantity of virus is many logs higher than what is estimated to cause infection in humans in vivo. Studies of the "sexual transmission bottleneck" estimate that a single founder virus or clone establishes infection in humans. Therefore the 'filling' of the reactivation-prone reservoir early in infection in macaques may not be modelling what is occurring in humans. This caveat needs to be discussed.

3) The authors conclude that peripheral blood measures of the viral reservoirs are insufficient to predict reactivation from latency. However, although there is not good quantitative correlation, the predictive value for the combination of treatment duration and DNA levels of interruption seems pretty good (R^2^=0.88). Thus, the authors should consider mentioning this in the Abstract and revising the Title.

4) It would be interesting to know whether proviral copy numbers per individual cell differ between the 4 dpi and 10 or 27 dpi treatment initiation settings. Although these viruses have evolved mechanisms to prevent super/coinfections these do occur and seem more likely in an environment where many cells are already infected. It would be interesting to know whether the same proviral copy numbers for the 4 dpi treatment corresponds to a larger number of latently infected cells than the later time points since this could potentially help to explain some findings.

5) Subsection “Measuring the frequency of SIV reactivation from latency”: what evidence exists to support the hypothesis that all proviruses contribute equally to HIV reactivation? The opposite hypothesis, that most proviruses have little-to-no contribution to viral recrudescence and a small minority are capable of reactivation and viral production, has much evidence to support it, including heterogeneity of the latent reservoir with respect to proviral integration site, accumulation of defective proviruses containing hypermutations / insertions / deletions, epigenetic phenomena contributing to transcription threshold, unique half-lives of T cell subsets and potential differences arising from local tissue microenvironments (tissue vs. peripheral blood) all strongly suggest that proviruses cannot be assumed to all possess the same potential for proviral reactivation.

6) Subsection “Measuring the frequency of SIV reactivation from latency”: doesn't this suggest an accumulation of defective proviruses over time, that increases the longer viral replication is allowed to proceed? The initial viral infection would be with a large population of replication-competent virions. Subsequent replication in vivo will give rise to defective proviruses in accordance with the error-prone nature of SIV RT. It is well known that the reservoir is established early in primary infection. Therefore, if ART is started within days of primary infection, the proportion of intact proviruses will be much higher than if ART is delayed and replication persists. It looks as though this hypothesis is addressed to some degree in the Discussion section however this is not included in this manuscript. The estimate of 80% of proviruses as genetically intact is considerably higher than what is observed in human infection in vivo (40-fold higher, per the Siliciano IPDA Nature paper). Do the authors think this marked difference between the SIV model and human HIV infection with regard to the proportion of intact proviruses may affect their results or make these findings challenging to apply to human infection?

---

## [Author Response]

Essential revisions:1) The manuscript focuses almost exclusively on the measurement of total proviral DNA, and does not directly take into account that in humans this measurement vastly over-estimates the reservoir size due to the accumulation of defective proviruses that functionally may not contribute to the reservoir at all. In the Discussion section of this work, the authors dismiss this phenomenon by pointing to an upcoming manuscript evaluating this in the SIV model and provide a very high frequency of intact proviruses (80%, section). No data or methods accompany this, and it is markedly different than what is seen in humans (2% of total proviruses are thought to be genetically intact) – which raises some concerns about the nature of this marked difference and the applicability of this model to the human HIV reservoir. This should be addressed in more detail or (better) substantiated by experimental data.

We thank the reviewers for this suggestion. The issue of replication competence is clearly one that needs better explanation, as it is touched upon by several of the comments (including Essential revision 6). We have now added a new methods section, and a new Results section and figure (new Figure 5) in order to explain the approach in more detail. We note that this data is reproduced from a publication that is now in press (Long et al., in press), as is explained in the manuscript:

Subsection “Single genome amplification and sequencing of full-length SIV genome”:

“Genomic DNA was purified from PBMC from SIV-infected rhesus macaques using QIAamp DNA Mini Kit (Qiagen). […] SGA PCR products were directly sequenced using BigDye Terminator Sanger Sequencing (Life Technologies) with the primers previously described (Lopker et al., 2016).”

Discussion section:

“However, near full length viral sequencing from animals treated on day 10 and day 27 revealed that the vast majority (>80%) of sequences were intact. […] Taken together, it is clear that the observed differences in reactivation frequency between animals treated on different days cannot be explained by different proportions of replication competent virus.”

In addition, we now address the differences in replication competence in HIV (point 6 below), citing recent work on the proportion of intact viral DNA in HIV and SIV.

2) The viral inoculum that gives rise to primary infection here is many orders of magnitude larger than what is thought to cause primary infection in humans (at least via sexual transmission), and this may affect the kinetics of reservoir establishment – making this a less-than-ideal model of human reservoirs. How much of the concept of 'early / rapid filling' of the reservoir shown here is an artefact of the non-human primate SIV infection model, in which many thousands of intact virions are used simultaneously for primary infection? This quantity of virus is many logs higher than what is estimated to cause infection in humans in vivo. Studies of the "sexual transmission bottleneck" estimate that a single founder virus or clone establishes infection in humans. Therefore the 'filling' of the reactivation-prone reservoir early in infection in macaques may not be modelling what is occurring in humans. This caveat needs to be discussed.

We thank the reviewers for suggesting further discussion of this point, as it is one that we feel is sometimes misunderstood. The monkeys are infected with a large dose of virus, and by a different route (intravenous), compared to sexual transmission of HIV. Therefore, it is worth asking ‘does this early virus drive earlier reservoir formation’? Considering the animals treated on day 4, for example, how much of the virus present at ART initiation might be the result of the initial high inoculum? The viral load increases around 10 fold per day in early infection (growth rate of the virus is around 2 day^-1^). Thus, broadly speaking we expect the viral load on day 4 to be around 10,000 fold higher than the viral load at infection. Even if we are off by several orders of magnitude, it is still clear that the initial inoculum does not play a big role in the total population of virus treated on day 4.

Another way to frame this is to ask, ‘do we have evidence for early establishment of the reservoir in HIV?’ We do not think that the exact timing of reservoir establishment in high-dose infected macaques will be precisely recapitulated in humans, but the idea of establishing the rebound competent reservoir early in the infection process is valid and has already been demonstrated in a limited cohort of infected humans. We would argue that the studies of Colby and colleagues (Colby et al., 2018) indicate that not only is the reservoir established extremely early in HIV, but that – as we find in SIV – the frequency of reactivation is almost as high as would be observed following treatment in chronic infection.

We think this is an important point to clarify, and have added a paragraph to the Discussion section:

“The initial inoculum of virus in our studies is several orders of magnitude higher than that estimated in HIV (Keele et al., 2008), and thus this may disproportionately contribute to the early establishment and potential saturation of the SIV RCR. However, since viral loads increase around 10-fold per day in early infection (Figure 1A,C,E), it is unlikely that this initial inoculum virus equates to >0.1% of virus present even at day 4 post-infection (the timing of earliest treatment). Thus, it seems unlikely that the higher inoculum in SIV infection could be a major contributor to the SIV reservoir measured later during treatment.”

3) The authors conclude that peripheral blood measures of the viral reservoirs are insufficient to predict reactivation from latency. However, although there is not good quantitative correlation, the predictive value for the combination of treatment duration and DNA levels of interruption seems pretty good (R^2^=0.88). Thus, the authors should consider mentioning this in the Abstract and revising the Title.

We thank the reviewers for pointing this out and have altered the Abstract and Title to reflect this. This is a good point, and we have changed the Title to “Predictors of SIV recrudescence following antiretroviral treatment interruption”.

4) It would be interesting to know whether proviral copy numbers per individual cell differ between the 4 dpi and 10 or 27 dpi treatment initiation settings. Although these viruses have evolved mechanisms to prevent super/coinfections these do occur and seem more likely in an environment where many cells are already infected. It would be interesting to know whether the same proviral copy numbers for the 4 dpi treatment corresponds to a larger number of latently infected cells than the later time points since this could potentially help to explain some findings.

This is an interesting question, which could potentially play some role in the relationship between proviral copy number and reactivation frequency. Unfortunately we do not have experimental data on the rate of coinfection in these animals, and to our knowledge this has not been studied in SIV (although studies in HIV suggest the rate is relatively low (Josejsson et al., 2011)).

Interestingly, it is not entirely clear that coinfection would change the relationship between DNA copies and frequency of reactivation in our system. That is, we detect the number and size ratio of different barcode sizes. If a cell harbours two different barcodes and both ‘wake up’ independently, then we would see and count the second barcode (unless the coinfection was with identical barcodes). Moreover, it is not clear whether the limitation on viral reactivation is ‘infected cells’ (and the probability of a cell waking up), or ‘DNA copies’ (and the probability of a DNA copy being integrated in a site where it might reactivate). Coinfection would largely be an issue in the former case, and much less so in the latter.

This seems a little too complex to go into in our study, however we have added a line in the Discussion section to state:

“Similarly, although coinfection of cells with multiple copies of HIV is thought to be low (Josejsson et al., 2011), rates of SIV coinfection of cells were not determined in our study and could play a role in determining the frequency of reactivation per DNA copy observed.”

5) Subsection “Measuring the frequency of SIV reactivation from latency”: what evidence exists to support the hypothesis that all proviruses contribute equally to HIV reactivation? The opposite hypothesis, that most proviruses have little-to-no contribution to viral recrudescence and a small minority are capable of reactivation and viral production, has much evidence to support it, including heterogeneity of the latent reservoir with respect to proviral integration site, accumulation of defective proviruses containing hypermutations / insertions / deletions, epigenetic phenomena contributing to transcription threshold, unique half-lives of T cell subsets and potential differences arising from local tissue microenvironments (tissue vs. peripheral blood) all strongly suggest that proviruses cannot be assumed to all possess the same potential for proviral reactivation.

We agree this was poorly worded and did not convey our intent. We intended to discuss whether the DNA measured in different animals contributed equally to reactivation – not to indicate that all DNA copies contributed equally. We have tried to reword this section to discuss whether DNA on average made a similar contribution (subsection “SIV DNA in PBMC and frequency of reactivation”).

“Although individual SIV genomes may vary greatly in their probability of reactivation (for example due to replication competence, integration site, or cell phenotype, activation state, and epigenetic or transcriptional blockades), if SIV DNA measured in the animals treated on different days had on average a similar probability of contributing to SIV reactivation from latency, then a doubling of DNA would equate to a doubling of reactivation (and we should expect a 1:1 correlation in this log:log plot).”

6) Subsection “Measuring the frequency of SIV reactivation from latency”: doesn't this suggest an accumulation of defective proviruses over time, that increases the longer viral replication is allowed to proceed? The initial viral infection would be with a large population of replication-competent virions. Subsequent replication in vivo will give rise to defective proviruses in accordance with the error-prone nature of SIV RT. It is well known that the reservoir is established early in primary infection. Therefore, if ART is started within days of primary infection, the proportion of intact proviruses will be much higher than if ART is delayed and replication persists. It looks as though this hypothesis is addressed to some degree in the Discussion section however this is not included in this manuscript. The estimate of 80% of proviruses as genetically intact is considerably higher than what is observed in human infection in vivo (40-fold higher, per the Siliciano IPDA Nature paper). Do the authors think this marked difference between the SIV model and human HIV infection with regard to the proportion of intact proviruses may affect their results or make these findings challenging to apply to human infection?

We agree that the issue of replication competence needed further explanation, and have added a section on our sequencing method, as well as expanding our discussion of replication competence in both SIV and HIV (the changes are detailed in the response to Q1 above).